# Sources of PM$_{2.5}$-Associated PAHs and n-alkanes in Changzhou China

**Ning Sun [1,\*], Xudong Li [1], Ye Ji [1], Hongying Huang [1], Zhaolian Ye [1,2,\*] and Zhuzi Zhao [1,\*]**

[1]   College of Chemistry and Environmental Engineering, Jiangsu University of Technology, Changzhou 213001, China; d17851080485@163.com (X.L.); jy1078523590@163.com (Y.J.); hhy@jsut.edu.cn (H.H.)

[2]   Collaborative Innovation Center of Atmospheric Environment and Equipment Technology, Jiangsu Key Laboratory of Atmospheric Environment Monitoring and Pollution Control, School of Environmental Sciences and Engineering, Nanjing University of Information Science and Technology, Nanjing 210044, China

\*   Correspondence: sn850206399@163.com (N.S.); bess_ye@jsut.edu.cn (Z.Y.); zhaozz@jsut.edu.cn (Z.Z.); Tel.: +86-519-86953064 (Z.Y. & Z.Z.)

**Abstract:** Polycyclic aromatic hydrocarbons (PAHs) and n-alkanes are important specific organic constituents in fine particulate matter (PM$_{2.5}$). Seventy-five PM$_{2.5}$ samples were collected in Spring Changzhou, to investigate the concentrations and sources of n-alkanes (C$_9$–C$_{40}$) and PAHs. The average concentrations of total PAHs ($\sum$PAHs) and n-alkanes ($\sum$n-alkanes) were $4.37 \pm 4.95$ ng/m$^3$ and $252.37 \pm 184.02$ ng/m$^3$, ranging from 0.43 to 22.22 ng/m$^3$ and 57.37 to 972.17 ng/m$^3$, respectively. The average concentrations of $\sum$n-alkanes and $\sum$PAHs were higher in severely polluted days (PM$_{2.5} \geq 150$ μg/m$^3$) in comparison to other days. Up to 85% of PAHs were four- and five-ring compounds, and the middle-chain-length n-alkanes (C$_{25}$–C$_{35}$) were the most abundant species (80.9%). The molecular distribution of n-alkanes was characterized by odd-number carbon predominance (carbon preference index, CPI > 1), with a maximum centered at C$_{27}$, C$_{29}$, and C$_{31}$ revealing a significant role of biogenic sources. Principal component analysis suggested that the biogenic sources that contributed the most to n-alkanes and PAHs were from coal combustion (46.3%), followed by biomass burning (16.0%), and vehicular exhaust (10.3%). The variation in the concentration of n-alkanes and PAHs from different air mass transports was not agreement with the change in PM$_{2.5}$ mass, indicating that regional transport had important impacts on the characterization of PM$_{2.5}$. The results of our study can provide useful information for evaluating the influence of anthropogenic and biogenic activities on organic matters (n-alkanes and PAHs).

**Keywords:** n-alkanes; PAHs; principal component analysis; backward trajectory analysis

## 1. Introduction

With the rapid development of industrialization and urbanization, the economy of China has progressed quickly in the recent years, but the economic boom has caused huge pressure on the environment at the same time [1]. Atmospheric particulate matter, with aerodynamic diameters equal to or less than 2.5 μm (PM$_{2.5}$), is considered to be the main contributor to heavy air pollution, and it has important effects on human health, atmospheric visibility, and climate [1–3]. The compositions of PM$_{2.5}$ are very complicated, including organic carbon (OC), elemental carbon (EC), trace elements, inorganic salts, and some organic species, such as n-alkanes and polycyclic aromatic hydrocarbons (PAHs) [4]. PAHs and n-alkanes are important components of organic matter, although they only contribute a minor proportion (<1%) to the total PM$_{2.5}$ mass [5]. They are ubiquitous in the atmosphere and are primarily produced by the incomplete combustion of fossil fuels. PAHs and n-alkanes are non-polar organic compounds and are considered to be intermediate/semi-volatile organic compounds (I/SVOCs), with a saturation concentration

(C*) of $10^3$–$10^6$ µg/m$^3$. Further, n-alkanes could provide insight into the origins and long-range transport of aerosols, for example, Wang et al. [6] highlighted the importance of organic tracers in achieving more-accurate source apportionment, by comparing positive matrix factorization runs with and without the inclusion of organics. Additionally, n-alkanes is a group of organic compounds in both the atmospheric gas phase and particle phase, which experience oxidative aging via gas-phase oxidation and heterogeneous reaction in the atmosphere [7,8]. The n-alkanes might be generated by both anthropogenic sources (e.g., fossil fuels combustion, vehicle emission) and biogenic sources (e.g., terrestrial plant wax, aquatic macrophytes) [9,10]. The homologue distribution of n-alkanes may indicate different pollution sources, e.g., low-carbon chain n-alkanes are mainly emitted by fossil fuels, while vegetative detritus are the main contributors to $\geq C_{29}$ odd-number carbon alkanes [11]. PAHs have attracted increasing concern, due to their carcinogenic and mutagenic properties [12–14]. Over the past few decades, a large number of studies have focused on the concentration, temporal and spatial variation, risk assessment, and source of PAHs in $PM_{2.5}$ [15,16]. Furthermore, several studies [17–19] discussed n-alkanes and PAHs at the same time, due to their similar characters (long-lived and stable in atmosphere) and formation process of incomplete combustion. Previous studies [9,20] have showed that the concentrations of n-alkanes and PAHs in the pollution period were one order of magnitude higher than those in the clean period; however, much less attention [21] has been given to comparing the characteristics and sources of n-alkanes or PAHs on polluted days with non-polluted days. Since specific organics in $PM_{2.5}$ can provide insights into valuable source signatures, it is very essential and helpful to gain comprehensive information on both n-alkanes and PAHs, to formulate effective control strategies for organic matter during different pollution days.

Changzhou is located in the Yangtze River Delta (YRD), with area of 4385 km$^2$, and it is a region with a developed economy in China. In recent years, with the rapid development of the industries, the problems of air pollution in Changzhou have become more serious. Although some researchers analyzed the characteristics and sources of water-soluble ions (WSIIs), organic carbon (OC), elementary carbon (EC), and heavy metals, in atmospheric $PM_{2.5}$ in Changzhou [22–26], less attention has been given to specific organic compounds (e.g., PAHs and n-alkanes). To address this gap, we performed chemical characterizations of daily $PM_{2.5}$ samples in Changzhou during spring, and conducted comprehensive analysis on 32 n-alkanes components ($C_9$–$C_{40}$) and 17 kinds of PAHs in different periods. The $PM_{2.5}$, OC, EC, and some other components were simultaneously determined. The possible sources were determined by diagnostic parameters and principal component analysis (PCA). This study tried to fill up new data of chemical components in $PM_{2.5}$, which would have important reference values for air pollution control in the Yangtze River Delta region.

## 2. Materials and Methods

### 2.1. Sampling Sites and Sample Collection

The sampling site is located on the nine-story platform of building in Jiangsu University of Technology (31.7° N, 119.9° E), approximately 30 m away from the ground. Details of the site and its surroundings were described in our previous work [4]. Quartz filters with 8 inches × 10 inches (QM-A, Whatman: Maidstone, UK) were used to collect $PM_{2.5}$ samples. Notably, filters should be prebaked at 500 °C for 4 h prior to sample collection in order to eliminate possible organic matters. Ultimately, a total of 75 $PM_{2.5}$ samples were collected by using high-volume $PM_{2.5}$ samplers (KB-1000, Qingdao Genstar Ltd., Qingdao, China) at a flow rate of 1.05 m$^3$/min from March to May in 2017. The samples were collected for 22 h, from 18:00 p.m. to 16:00 p.m. of the next day. The filters were weighed with a precision balance (accuracy of 0.01 mg) before and after sampling to obtain the mass of samples, then stored in aluminum foil packages separately and placed in a freezer at −20 °C until analysis. During the sampling period, the weather parameters such as temperature, wind speed, relative humidity and atmospheric pressure were recorded.

In order to distinguish the pollution characteristics and sources during different periods, in this article, the data were separated into non-polluted days ($PM_{2.5} \leq 75 \, \mu g/m^3$), polluted days ($150 > PM_{2.5} > 75 \, \mu g/m^3$) and severely polluted days (daily $PM_{2.5} \geq 150 \, \mu g/m^3$) based on the national daily air quality standard in China (NAAQS) grade II of $PM_{2.5}$, as suggested by Li [27].

### 2.2. Sample Analysis

#### 2.2.1. Analysis of OC, EC and Water-Soluble Ions

The concentration of OC and EC were measured by a thermal/optical carbon analyzer (model 2001A, Desert Research Institute, Reno, NV, USA) following a thermal/optical reflectance (TOR) protocol proposed by Chow et al. [28], and the details of the program were described in our previous work [4,29].

The water-soluble fractions in one-sixteenth of the filter samples were extracted with 50 mL ultrapure water under ultra-sonication for 45 min. The liquid extract was filtered with PTFE microporous membrane (pore size of 0.45 μm) to remove the particles and filter debris. Concentrations of the water-soluble ions in the aqueous extract, including three anions ($Cl^-$, $NO_3^-$, $SO_4^{2-}$) and five cations ($Na^+$, $NH_4^+$, $K^+$, $Mg^{2+}$, $Ca^{2+}$) were measured by the ion chromatograph (IC, Aquion, Thermo Fisher Scientific Inc., Waltham, MA, USA).

#### 2.2.2. Analysis of Polycyclic Aromatic Hydrocarbons

An Agilent 7890A/5975C system (Agilent: Santa Clara, CA, USA) with in-injection port thermal desorption gas chromatography/mass spectrometry (TD-GC-MS) method was used to quantify PAHs on the filter samples [30]. In brief, each filter was cut into small pieces (~0.5 cm²), and then put into a TD glass tube (78 mm long, inner diameter (ID) 4 mm, outer diameter (OD) 6.35 mm) in order to inject into the instrument. Thermal desorption was used for injection in a splitless mode; the initial injector temperature was held at 50 °C, then increased to 275 °C for desorption (8–9 min). GC used a capillary column (HP-5 ms, 50 m × 0.332 mm × 0.17 μm) to separate organic compounds. The electron ionization (EI) was set at 70 eV and the selected ion mode (SIM) was used for quantification. Over temperature was programmed from 30 °C (hold for 2 min) to 120 °C at 10 °C/min, ramped to 325 °C at 8 °C/min, and finally held at 325 °C for 20 min.

Seventeen PAH compounds were detected, including acenaphthene (Ace), fluorene (Flu), phenanthrene (Phe), anthracene (Ant), fluoranthene (Fla), pyrene (Pyr), benzo[a]anthracene (BaA), chrysene (Chry), benzo[b]fluoranthene (BbF), benzo[k]fluoranthene (BkF), benzo[a]-fluoranthene (BaF), benzo[e]pyrene (BeP), benzo[a]pyrene (BaP), perylene (Per), indeno[1,2,3-cd]pyrene (InP), benzo[ghi]perylene (BghiP) and dibenzo[a,h]anthracene (DahA). The total PAHs concentration ($\sum$PAHs) was calculated as the sum of mass of 17 individual PAHs.

#### 2.2.3. Analysis of n-alkanes

Further, n-alkanes of 1/8th pieces of filter were extracted with 25 mL n-hexane/dichloromethane (1:2, *v/v*) mixtures by accelerated solvent extractor (ThermoFisher, ASE 350, USA). The extracts were then filtered through a 0.45 μm PTFE syringe filter and transferred to a round-bottom flask. The extracts were concentrated to approximately 5 mL at 35 °C (water bath) by a rotary evaporator (EYELA, Tokyo, Japan), and then transferred to glass chromatography columns. Certain volume of n-hexane (~20 mL) passed through column to elute n-alkanes. Subsequently, the samples were condensed almost to dryness (less than 1 mL) with a stream of nitrogen gas. The remaining solution was transferred to a 1.5 mL brown vial. Then, 25 μL internal standard (dodecyl benzene, 20 mg/L) was added and final volume was fixed to 1.0 mL with n-hexane for quantitative analysis by a GC-MS (Agilent 7890-7000B GC-5975C MS, Agilent: Santa Clara, CA, USA) with DB-5MS capillary column (30 m × 0.25 mm × 0.25 μm). The electron energy was set to 70 eV while the ion source temperature was 230 °C. The temperature operation program was as follows: the initial column temperature was 50 °C (held for 5 min), then elevated at 15 °C/min to 160 °C (held for 5 min), and finally ramped up to 300 °C at 6 °C/min (held for 15 min).

*2.3. Quality Assurance and Quality Control (QA/QC)*

In order to reduce measure errors, rigid quality control processes were performed in the study. Blank filters were analyzed to correct sample concentrations. Method detection limits (MDLs) were defined as three times the standard deviations of the blank sample. The MDLs for OC and EC were 0.6 µg/m$^3$ and 0.2 µg/m$^3$, respectively. The MDLs of water-soluble ions were 0.01–0.05 µg/m$^3$ for anions and 0.01–0.08 µg/m$^3$ for cations. Moreover, the relative standard deviations of OC, EC and WSIIs were found to be less than 5%. In addition, during the analysis of PAHs and n-alkanes, one sample was randomly chosen from each group of ten for a replicate analysis. The standard deviation for duplicate analysis of each compound was below 15%. Additionally, 500 ng of the standards was spiked onto a blank filter and analyzed to determine their recoveries, and the average recovery of triplicate experiments ranged from 75% to 115%. The MDLs of each PAHs species and homologues for n-alkanes are presented in Table S2.

## 3. Results and Discussions

*3.1. PM$_{2.5}$ and Its Constituents*

Table 1 lists the average mass concentration and standard deviation of PM$_{2.5}$ and its main chemical components under different polluted levels. The average daily concentration of PM$_{2.5}$ in Changzhou, during the sampling period, was 101.97 ± 35.45 µg/m$^3$, which was close to our previous study in Changzhou during spring in 2016 (106.0 ± 24.4 µg/m$^3$) [4], indicating that the air quality in Changzhou has not improved significantly, and exceeded the second-grade national ambient air quality standard of 75 µg/m$^3$ (GB3095-2012). Overall, the absolute concentration of components in PM$_{2.5}$ exhibited the highest level on severely polluted days, followed by polluted days. The average concentrations of OC and EC were 14.39 and 5.76 µg/m$^3$, accounting for 14.1% and 5.6% of PM$_{2.5}$ mass, respectively. Further calculation showed that the average OC/EC ratio during the whole period was 2.5. The mass concentrations of total water-soluble ions (∑WSIIs) were 37.72 ± 15.00 µg/m$^3$, accounting for 38.5 ± 10.6% of PM$_{2.5}$. In addition, SNA (NO$_3^-$, SO$_4^{2-}$, and NH$_4^+$) were the dominate components of water-soluble ions, with a percentage of 84.5% in WSIIs.

For different typical days, the proportions of OC in PM$_{2.5}$ decreased from 15.3% (non-polluted), 13.9% (polluted), and 12.7% (severely polluted), also that for EC decreased from 6.5% (non-polluted), 5.9% (polluted), and 4.9% (severely polluted). Interestingly, we also found that the SNA showed a downward trend from non-polluted to severely polluted days. This was a different pattern from the previous studies in Changzhou winter [31,32], or other cities [1,2,10], which generally showed that SNA proportions increased in polluted periods. We speculated that in spring, the PM$_{2.5}$ concentrations might be somewhat impacted by dust storms, and thus this caused a relative decrease in the percentages of SNA and carbonaceous species. Moreover, this conclusion could be manifested by the significantly increased dust in PM$_{2.5}$ from non-polluted days (11.2%) to severely polluted days (16.2%) (Figure 1). Another survey [33] also observed the important transport of a dust storm from the north of China for PM$_{2.5}$ in Shanghai spring.

In general, the average total measured n-alkanes concentration (∑n-alkanes) was 486.22 ± 260.44 ng/m$^3$ on severely polluted days, 241.43 ± 178.51 ng/m$^3$ on polluted days, and 200.99 ± 113.97 ng/m$^3$ on non-polluted days, respectively. The concentration of n-alkanes in this study was comparable to that (227.6 ng/m$^3$) in Shanghai [5]. The ∑n-alkanes during severely polluted days was approximately 2–3 times that of other days. Additionally, the mean concentration ratio between severely polluted days and non-polluted days of ∑PAHs was 4.1 (9.59 ng/m$^3$ vs. 2.36 ng/m$^3$). For n-alkanes, the proportion of ∑n-alkanes in PM$_{2.5}$ was 0.37%, 0.23%, and 0.31% on non-polluted, polluted, and severely polluted days, respectively. This trend showed that the dust had a certain influence on the source of PM$_{2.5}$, and this is described in Section 3.5, as the high concentration of n-alkanes with low PM$_{2.5}$ in cluster 2 from the northwest cities of China. For the PAHs proportions of PM$_{2.5}$, they decreased from 0.006% to 0.004% on severely polluted and non-polluted days, respectively,

indicating that with the aggravation of pollution, PAHs had a slightly higher role in $PM_{2.5}$. The further analysis of n-alkanes and PAHs was conducted in Sections 3.2 and 3.3.

**Table 1.** Mean concentration (±standard deviation) of $PM_{2.5}$ and its major constituents in different pollution periods.

| Constituents | Severely Polluted (n = 8) | Polluted (n = 48) | Non-Polluted (n = 19) | All (n = 75) |
|---|---|---|---|---|
| | Average ± Std | Average ± Std | Average ± Std | Average ± Std |
| $PM_{2.5}$, carbonaceous contents ($\mu g/m^3$) | | | | |
| $PM_{2.5}$ | 159.53 ± 9.36 | 107.85 ± 19.33 | 56.42 ± 10.59 | 101.97 ± 35.45 |
| EC | 7.76 ± 1.55 | 6.36 ± 1.86 | 3.67 ± 1.08 | 5.76 ± 2.04 |
| OC | 20.24 ± 5.82 | 15.02 ± 4.08 | 8.64 ± 3.33 | 14.39 ± 5.96 |
| Water-soluble ions ($\mu g/m^3$) | | | | |
| $NO_3^-$ | 25.22 ± 9.10 | 15.77 ± 7.96 | 8.30 ± 3.74 | 15.10 ± 9.05 |
| $NH_4^+$ | 9.63 ± 2.09 | 7.46 ± 2.46 | 5.61 ± 1.39 | 7.27 ± 2.51 |
| $SO_4^{2-}$ | 12.33 ± 2.82 | 10.27 ± 3.60 | 7.96 ± 1.68 | 9.93 ± 3.39 |
| $Mg^{2+}$ | 0.17 ± 0.06 | 0.09 ± 0.05 | 0.03 ± 0.03 | 0.09 ± 0.06 |
| $Ca^{2+}$ | 4.67 ± 1.35 | 2.85 ± 1.64 | 1.15 ± 0.74 | 2.62 ± 1.72 |
| $K^+$ | 0.61 ± 0.28 | 0.39 ± 0.20 | 0.24 ± 0.21 | 0.40 ± 0.27 |
| $Na^+$ | 1.35 ± 0.98 | 0.81 ± 0.42 | 0.75 ± 0.37 | 0.86 ± 0.50 |
| $Cl^-$ | 3.82 ± 2.48 | 1.78 ± 1.46 | 1.04 ± 1.05 | 1.88 ± 1.75 |
| SNA | 47.18 ± 12.59 | 33.50 ± 12.59 | 21.87 ± 6.00 | 32.20 ± 13.31 |
| $\sum$WSIIs | 57.80 ± 14.58 | 39.42 ± 12.98 | 25.09 ± 6.92 | 37.72 ± 15.00 |
| Organic compounds ($ng/m^3$) | | | | |
| $\sum$n-alkanes | 486.22 ± 260.44 | 241.43 ± 178.51 | 200.99 ± 113.97 | 252.37 ± 184.02 |
| $\sum$PAHs | 9.59 ± 8.28 | 4.30 ± 4.44 | 2.36 ± 1.60 | 4.37 ± 4.95 |
| Percentage (%) | | | | |
| $EC/PM_{2.5}$ | 4.9 ± 1.0 | 5.9 ± 1.8 | 6.5 ± 1.1 | 5.6 ± 1.6 |
| $OC/PM_{2.5}$ | 12.7 ± 3.7 | 13.9 ± 4.1 | 15.3 ± 4.5 | 14.1 ± 4.2 |
| SNA/$\sum$WSIIs | 81.6 ± 5.7 | 85.0 ± 8.2 | 87.2 ± 4.3 | 84.5 ± 7.3 |
| $\sum$WSIIs/$PM_{2.5}$ | 36.2 ± 8.8 | 36.6 ± 9.3 | 44.5 ± 12.3 | 38.5 ± 10.6 |
| $\sum$n-alkanes/$PM_{2.5}$ | 0.31 ± 0.17 | 0.23 ± 0.16 | 0.37 ± 0.25 | 0.27 ± 0.20 |
| $\sum$PAHs/$PM_{2.5}$ | 0.006 ± 0.005 | 0.004 ± 0.004 | 0.004 ± 0.003 | 0.004 ± 0.004 |

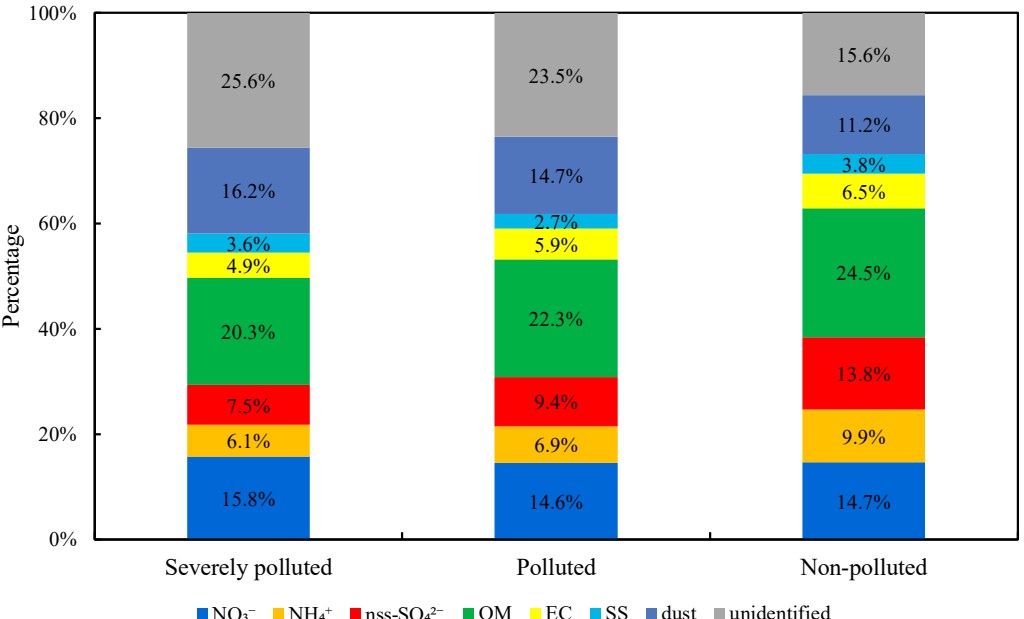

**Figure 1.** Percentage of chemical components attributable to $PM_{2.5}$ mass in three pollution days.

The time series of the concentrations of $PM_{2.5}$, $\sum$n-alkanes, and $\sum$PAHs in Changzhou are shown in Figure 2. Of 75 samples, polluted days (48 days) accounted for 64%, followed

by non-polluted (19 days, 25%) and severely polluted (8 days, 11%) days. The concentrations of $PM_{2.5}$, $\sum$n-alkanes, and $\sum$PAHs fluctuated dramatically, from 38.5 to 177.2 $\mu g/m^3$, 57.37 to 972.17 $ng/m^3$, and 0.43 to 22.22 $ng/m^3$, respectively. A weak positive relationship was found between $PM_{2.5}$ and $\sum$PAHs, due to the fact that combustion and other primary sources created fine particles carrying some PAHs compounds [34]. However, the concentration of $\sum$PAHs did not always increase with increasing $PM_{2.5}$ concentrations, since PAHs have long lifetimes (4–8 days) under ambient conditions [35].

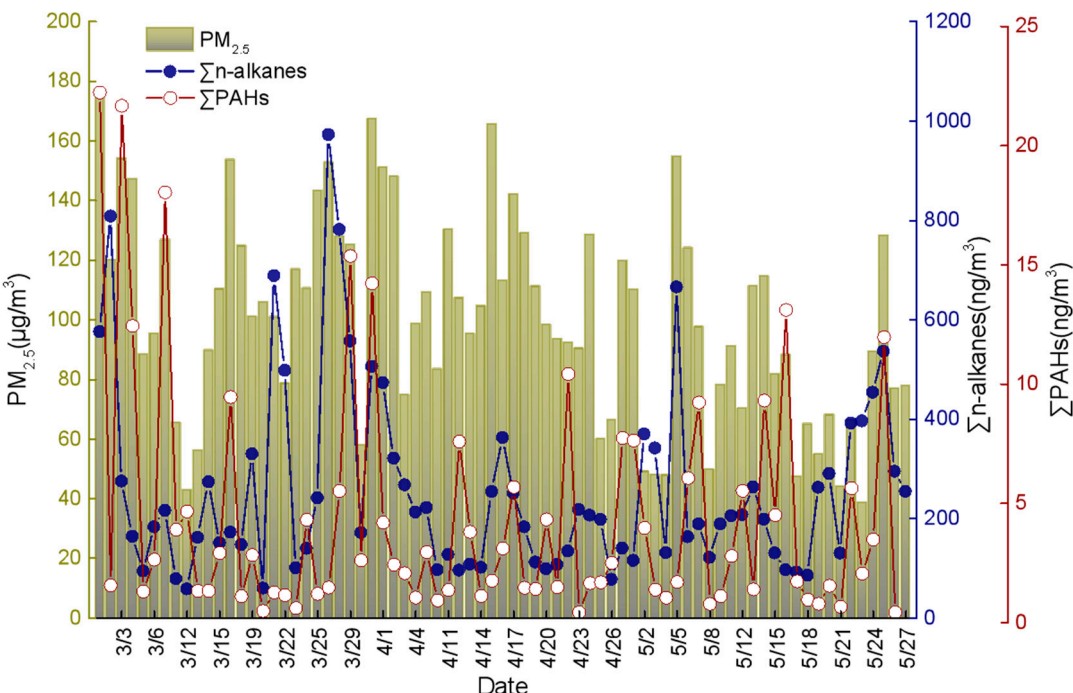

**Figure 2.** Time series of concentrations of $PM_{2.5}$, $\sum$n-alkanes and $\sum$PAHs in spring in Changzhou.

The chemical mass closures of $PM_{2.5}$ during three stages were calculated and presented in Figure 1. For mass reconstruction, seven components [36] were considered, i.e., (1) organic matter (OM), calculated as OC × 1.6; (2) EC; (3) non-sea-salt sulphate (nss-$SO_4^{2-}$), evaluated as $SO_4^{2-}$ − 0.231 × $Na^+$; (4) nitrate ($NO_3^-$); (5) ammonium ($NH_4^+$); (6) sea salt (SS), estimated as $Cl^-$ + 1.4486 $Na^+$; and (7) dust (($Ca^{2+}$ − 0.038 × $Na^+$) × 5.6). About 74–85% of $PM_{2.5}$ mass was identified. From Figure 1, it was evident that SNA were the most-abundant components of $PM_{2.5}$, accounting for 29.4%, 30.9%, and 38.4% of $PM_{2.5}$ mass in severely polluted, polluted, and non-polluted days, respectively. Sea salts comprised a relatively small percentage (2–4%). For more details, the percentages of $NO_3^-$ and $NH_4^+$ showed a slight change in the three pollution levels, but the mass of nss-$SO_4^{2-}$, which was attributed to the chemical reactions of $SO_2$ in the atmosphere, caused by fossil fuel combustion and direct emission of $SO_3$ from diesel engines [2], had an obvious percentage change from severely polluted (7.5%) to non-polluted (13.8%) days, showing the higher contribution of fossil fuel combustion in non-polluted days than other days. In addition, OM was the second-largest component during the three periods. With the increase in $PM_{2.5}$ concentrations, the relative abundance of OM decreased from 24.5% in the non-polluted period to 20.3% on the severely polluted days. The proportion of dust showed a marked decline from the severely polluted (16.2%) to non-polluted period (11.2%). Additionally, the proportions of EC, $NH_4^+$, and the SS had a slight change among the three stages. The percentage contribution of undetermined components showed a change under three pollution levels, with the maximum (25.6%) on severely polluted days and the minimum (15.6%) on non-polluted days. The unidentified mass might be associated with retained water, volatilization losses, tracer organics, as well as trace minerals [36].

### 3.2. Mass Concentration and Distribution of n-alkanes

The average concentrations of ∑n-alkanes on different polluted days are presented in Figure 3. As can be observed, the middle-chain-length n-alkanes ($C_{25}$–$C_{35}$) were the most abundant, accounting for, on average, 80.9% of the ∑n-alkanes. The carbon maximum number ($C_{max}$) was defined as the carbon number with the maximum concentration among the homologues of n-alkanes, which can provide an approach to assess anthropogenic versus biogenic source contributions. In general, high-$C_{max}$ n-alkanes come from biogenic sources and low-$C_{max}$ n-alkanes from anthropogenic sources [9,37]. The predominance of low carbon number ($C_{11}$–$C_{24}$) indicated significant anthropogenic inputs from increased vehicular emissions. Notably, fossil fuel combustion was characterized by the release of $C_{22}$–$C_{25}$ n-alkanes, while particulate abrasion products from leaf surfaces were characterized by the predominance of >$C_{29}$ odd n-alkanes. Perrone et al. [38] found that $C_{21}$–$C_{22}$ n-alkanes were predominantly emitted from diesel engine exhausts.

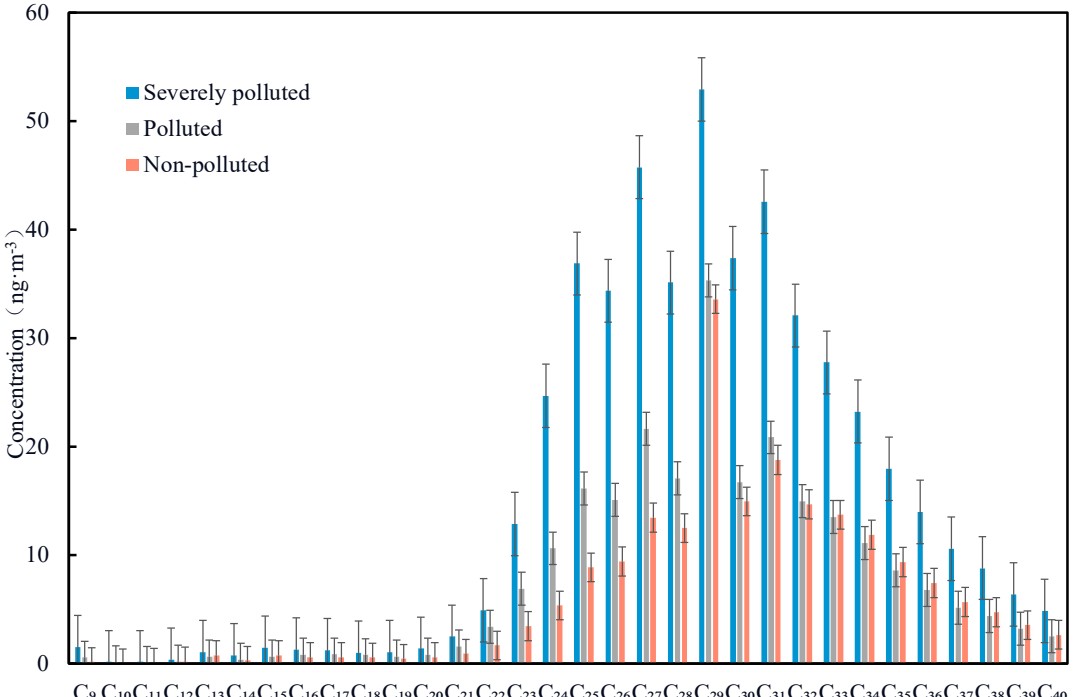

**Figure 3.** Carbon number distribution patterns for n-alkanes in different polluted days.

From Figure 3, it can be found that the most-abundant congener of n-alkanes was $C_{29}$ in the three stages. A study from Beijing [37] also showed that high-$C_{max}$ n-alkanes (such as $C_{27}$, $C_{29}$, and $C_{31}$) peaked in spring, due to great contribution from high plant wax. The average $C_{29}$ concentration on severely polluted days was 1.50 and 1.58 times higher than that on polluted and non-polluted days, respectively. The next abundant carbon-numbered congeners was $C_{27}$ on severely polluted and polluted days, and $C_{31}$ on non-polluted days, respectively. In total, an odd high-carbon number preference indicated large contributions from biogenic sources, which was in good agreement with the concentration distributions of n-alkane in Guangzhou in autumn [39].

### 3.3. PAHs Mass Concentration and Distribution of Different Rings

The average concentrations of the 17 individual PAHs, and the distribution of different ring numbers of PAHs in the three pollution levels, are presented in Figure 4. It can be observed from Figure 4a that Chry and BkF were the two most-abundant PAH species, followed by Pyr, BaA, Fla, Phe, and BaP, during the whole period. Among those 17 PAHs, the sum of the carcinogenic PAHs (CANPAHs = BaA + BbF + BkF + BaP + DahA + InP) accounted for a relatively higher proportion (46.3%) of ∑PAHs concentrations. Special

attention should be given to PAH, due to its relatively strong carcinogenic effect, regardless of lower $\sum$PAHs concentrations [40]. The sum of the combustion-derived PAHs concentrations (COMPAHs, including Flu, Pyr, Chry, BbF, BkF, BaA, BeP, BaP, InP, and BghiP) occupied 74.9% of the total PAHs mass in $PM_{2.5}$.

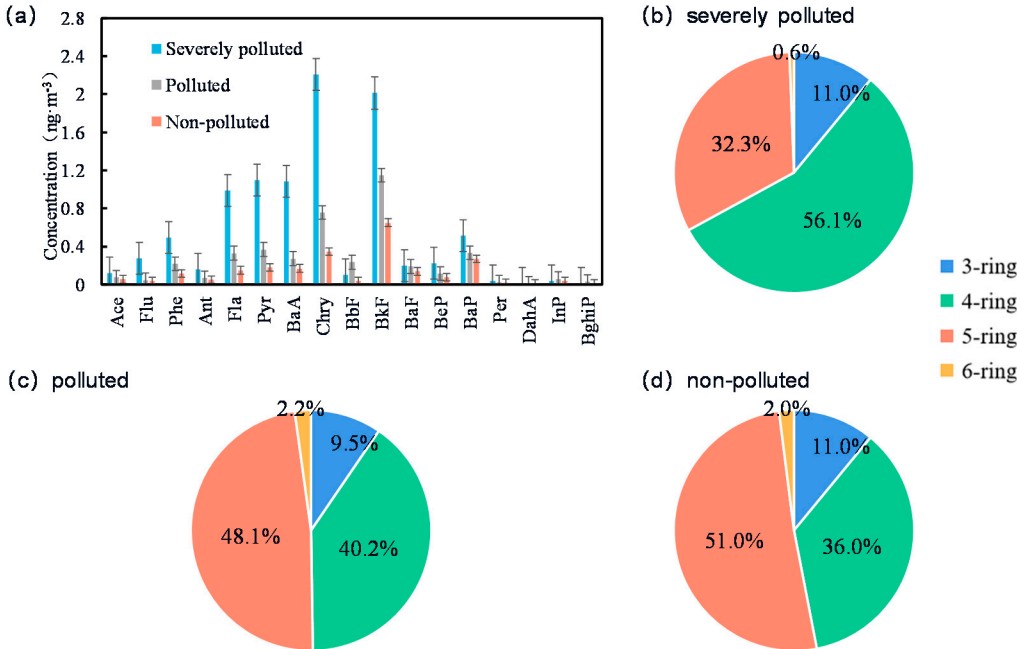

**Figure 4.** (**a**) Monomer average concentration and (**b**–**d**) distribution of different ring number of 17 PAHs in $PM_{2.5}$ in different polluted days.

The 17 PAHs evaluated can be classified into the following four groups: three-ring (Ace, Flu, Phe, Ant), four-ring (Fla, Pyr, BaA, Chry), five-ring (BbF, BkF, BaF, BeP, BaP, Per, DahA), and six-ring (InP, BghiP). As observed from Figure 4b–d, the relative contributions of PAHs with different rings varied with the pollution level. Four-ring and five-ring PAHs were the dominate components in total PAHs, and four-ring, on severely polluted days, contributed higher than that in the other two conditions. As we know, the four-ring PAHs mainly came from low-temperature combustion (i.e., coal and biomass combustion), while five-to-six-ring PAHs were mainly produced from high-temperature combustion processes, such as vehicle emissions [41,42]. In addition, high-molecular-weight PAHs, such as BkF and InP, were the emission from diesel-powered vehicles [43]. So, it can be inferred that the main sources of PAHs on severely polluted days were coal/biomass combustion, while vehicle emission contributed more on non-polluted days. The proportion of three-ring PAHs was relatively stable (9.0–11.0%) among the three periods. As we know [42], PAHs with two-to-three rings were estimated to be preferentially in the gas phase, due to its easy volatilization at above 25 °C and degradation under strong ultraviolet radiation; as a result, the concentration in the particle phase was low in this study. Six-ring PAHs contributed the minimum and the proportion of BghiP was slightly higher than that of InP.

### 3.4. Source Identification of n-alkanes and PAHs

#### 3.4.1. Source Identification of n-alkanes by Diagnostic Parameters

The diagnostic ratios of some organic indicators can provide additional information to identify source emissions. The carbon preference index (CPI), defined as the concentration ratio of odd-to-even carbon number n-alkanes, was a useful index to assess the relative contributions of biogenic and anthropogenic emissions to n-alkanes [37,44]. In this study, we define CPI as the average CPI for the whole range of n-alkanes ($C_9$–$C_{40}$), while CPI1 only denotes petrogenic n-alkanes ($C_9$–$C_{25}$), and CPI2 represents biologic n-alkanes

($C_{26}$–$C_{40}$). Furthermore, plant wax n-alkanes are characterized by a higher odd-number carbon ($C_{25}$–$C_{33}$), while n-alkanes from fossil fuel combustion show no significant carbon number predominance, and, thus, biogenic n-alkanes should have high CPI values, whereas anthropogenic n-alkanes should have CPI values close to one.

Table 2 lists the calculated values for CPI, CPI1, and CPI2 of n-alkanes, and Table 3 summarizes the previous results of CPI of n-alkanes. As shown in Table 2, the value of CPI ranged from 0.93 to 4.05, with an average of 1.35, implying mixed influences from both biogenic and anthropogenic activities. Moreover, the values in our finding were comparable to those from other cities listed in Table 3, and were within the range measured for urban environments (1.0–2.0), due to certain contributions from anthropogenic emissions. Overall, CPI has no significant difference in the three pollution levels, with weaker odd/even predominance under severely polluted days (CPI = 1.18) in comparison with non-polluted days (CPI = 1.39), indicating anthropogenic emissions from vehicular and fossil fuels probably contributed more to n-alkanes on severely polluted days. CPI1 ranged from 1.22 to 2.68 (average of 1.83), and the fact that all of the mean CPI1 values were greater than unity shows that anthropogenic sources affected the n-alkanes for the entire study. The CPI2 values over one (1.26) also indicated a certain contribution of other non-anthropogenic sources for this range of n-alkanes. All in all, n-alkanes were contributed to and impacted by both biogenic and anthropogenic activities.

**Table 2.** CPI of n-alkanes associated with $PM_{2.5}$ during different polluted days.

| | Severely Polluted | | Polluted | | Non-Polluted | | All | |
|---|---|---|---|---|---|---|---|---|
| | Mean | Range | Mean | Range | Mean | Range | Mean | Range |
| CPI | $1.18 \pm 0.12$ | 1.08–1.42 | $1.36 \pm 0.33$ | 0.93–2.55 | $1.39 \pm 0.67$ | 0.93–4.05 | $1.35 \pm 0.43$ | 0.93–4.05 |
| CPI1 | $1.84 \pm 0.53$ | 1.35–2.65 | $1.82 \pm 0.35$ | 1.22–2.68 | $1.84 \pm 0.23$ | 1.43–2.19 | $1.83 \pm 0.34$ | 1.22–2.68 |
| CPI2 | $1.05 \pm 0.15$ | 0.88–1.35 | $1.27 \pm 0.37$ | 0.84–2.57 | $1.31 \pm 0.74$ | 0.66–4.22 | $1.26 \pm 0.48$ | 0.66–4.22 |

**Table 3.** Comparison of CPI in our findings with values reported in other cities in China.

| Cities | Sampling Period | CPI | | | Main Source | Ref. |
|---|---|---|---|---|---|---|
| | | CPI | CPI1 | CPI2 | | |
| Zhengzhou | 2015, Spring | 1.3 | 1.1 | 1.4 | anthropogenic emission | [10] |
| Chongming Island | 2018, Summer | 2.2 | 0.74 | 2.9 | fossil fuel combustion | [20] |
| Shanghai | 2013, Spring | 1.5 | 1.4 | 1.5 | anthropogenic sources | [36] |
| Taian | 2016–2017 | 1.83 | - | - | biogenic sources | [45] |
| Xiamen | 2013, Winter | 0.96 | 0.86 | 0.99 | anthropogenic sources | [46] |
| | 2013, Summer | 1.27 | 1.51 | 1.11 | biogenic sources | [46] |
| Beijing | 2013, Autumn | 0.99 | 1.16 | 0.82 | petroleum, diesel and gasoline emissions | [47] |
| Shanghai | 2015, Spring | 1.85 | 1.66 | 1.91 | biogenic sources | [48] |
| Nanjing | 2015, Spring | 1.78 | 1.57 | 1.94 | biogenic sources | [48] |
| Ningbo | 2015, Spring | 1.57 | 1.93 | 1.49 | anthropogenic sources | [48] |
| Tianjin | 2007, Spring | 1.29 | 1.28 | 1.30 | anthropogenic sources | [49] |

### 3.4.2. Correlations between n-alkanes, PAHs and Other Chemical Species

Table 4 lists the correlations between $\sum$n-alkanes, $\sum$PAHs, and some chemical species in $PM_{2.5}$. It was shown that $\sum$PAHs had a moderate correlation with OC (r = 0.251, $p < 0.05$) and a strong correlation with EC (r = 0.640, $p < 0.01$), suggesting that PAHs mainly originated from primary sources, such as coal combustion, biomass burning, and vehicle exhaust. Furthermore, a weak positive correlation between $\sum$PAHs with $NO_3^-$ (r = 0.352, $p < 0.01$) indicated that the secondary reaction of NOx from vehicle exhaust may be a source of PAHs [1,2]. Also, $SO_4^{2-}$, as a secondary source for which a photochemical reaction may occur for $SO_2$ from coal combustion [1,2], also showed a relatively weak correlation with n-alkanes (r = 0.252). In addition, both $\sum$n-alkanes and $\sum$PAHs had a positive correlation with $K^+$ (a tracer of biomass burning) and $Cl^-$ (derived from sea salt

and coal combustion) [36], indicating that biomass burning and coal combustion were also important sources. A positive correlation was also found amongst n-alkanes and $Mg^{2+}$ (r = 0.244), and PAHs with $Mg^{2+}$ (r = 0.336) and $Ca^{2+}$ (r = 0.330), meaning that dust was an indispensable source.

**Table 4.** Correlations between n-alkanes, PAHs and other chemical species in $PM_{2.5}$.

| | OC | EC | $NO_3^-$ | $SO_4^{2-}$ | $NH_4^+$ | $Na^+$ | $K^+$ | $Mg^{2+}$ | $Ca^{2+}$ | $Cl^-$ | $\sum$n-alkanes | $\sum$PAHs |
|---|---|---|---|---|---|---|---|---|---|---|---|---|
| OC | 1 | | | | | | | | | | | |
| EC | 0.848 ** | 1 | | | | | | | | | | |
| $NO_3^-$ | 0.545 ** | 0.421 ** | 1 | | | | | | | | | |
| $SO_4^{2-}$ | 0.255 * | 0.315 ** | 0.568 ** | 1 | | | | | | | | |
| $NH_4^+$ | 0.396 ** | 0.288 * | 0.934 ** | 0.738 ** | 1 | | | | | | | |
| $Na^+$ | 0.169 | 0.319 ** | 0.017 | 0.128 | −0.026 | 1 | | | | | | |
| $K^+$ | 0.574 ** | 0.503 ** | 0.375 ** | 0.326 ** | 0.328 * | 0.590 ** | 1 | | | | | |
| $Mg^{2+}$ | 0.537 ** | 0.648 ** | 0.349 ** | 0.217 | 0.246 * | 0.534 ** | 0.560 ** | 1 | | | | |
| $Ca^{2+}$ | 0.511 ** | 0.656 ** | 0.194 | 0.109 | 0.039 | 0.407 ** | 0.342 ** | 0.754 ** | 1 | | | |
| $Cl^-$ | 0.331 ** | 0.266 * | 0.598 ** | 0.30 ** | 0.623 ** | 0.509 ** | 0.547 ** | 0.491 ** | 0.202 | 1 | | |
| $\sum$n-alkanes | 0.160 | 0.120 | 0.143 | 0.252 * | 0.192 | 0.156 | 0.294 * | 0.244 * | 0.211 | 0.255 * | 1 | |
| $\sum$PAHs | 0.251 * | 0.640 ** | 0.352 ** | 0.092 | 0.275 * | 0.286 * | 0.265 * | 0.336 ** | 0.330 ** | 0.409 ** | 0.133 | 1 |

** All the particle samples are significant at $p < 0.01$, respectively; * all the particle samples are significant at $p < 0.05$, respectively.

In summary, we can conclude that PAHs may come from coal combustion, biomass burning, and vehicle exhaust, while the source of n-alkanes was probably coal combustion. Further discussion on the sources of n-alkanes and PAHs was presented in Sections 3.4.3 and 3.4.4.

### 3.4.3. Source Identification of PAHs by PCA

While the molecular parameters could provide some information for exploring the source of pollutants, principal component analysis (PCA) was widely used for quantifying the contribution of each source [47]. In this study, PCA analysis was performed by SPSS 20.0 statistical analysis software. Principal component factors having eigenvalues >1 and factor loadings >0.5 were taken into consideration.

Three principal components (PCs) explained 72.6% of the total variance in PAHs (Table 5). The first component (PC1) accounted for 46.3%, and was mostly associated with low-molecular-weight PAHs (three-to-four-ring PAHs), such as Ace, Phe, Ant, Fla, Pyr, and Chr, which had been identified as markers of coal combustion [40,50–52]. Thus, PC1 was characterized as coal combustion sources. The second component (PC2) was enriched in BeP, BaP, and Per, accounting for 16.0% of the total variance in PAHs. Those components provided a similar profile for biomass burning, and BaP was a good tracer for biomass burning [10], so PC2 was identified as biomass burning. The third component (PC3), with 10.3% of the total variance, was associated with InP and BghiP. InP and BghiP are typical markers of traffic emissions, with InP being a marker of diesel engine emission, and BghiP of gasoline engine emission [34,51–53], so PC3 was linked with vehicle exhaust.

**Table 5.** Principal component analysis of PAHs in $PM_{2.5}$.

| Compounds | PC1 | PC2 | PC3 |
|---|---|---|---|
| Ace | **0.844** | −0.278 | 0.124 |
| Flu | 0.566 | −0.282 | −0.122 |
| Phe | **0.916** | −0.332 | −0.005 |
| Ant | **0.956** | −0.204 | 0.028 |
| Fla | **0.904** | −0.379 | −0.059 |
| Pyr | **0.929** | −0.329 | −0.068 |
| BaA | 0.785 | −0.320 | −0.047 |
| Chr | **0.860** | −0.182 | −0.015 |
| BbF | 0.323 | 0.197 | 0.335 |

**Table 5.** *Cont.*

| Compounds | PC1 | PC2 | PC3 |
|:---:|:---:|:---:|:---:|
| BkF | 0.744 | 0.489 | −0.108 |
| BaF | 0.540 | 0.390 | 0.350 |
| BeP | 0.612 | **0.641** | −0.165 |
| BaP | 0.570 | **0.769** | −0.034 |
| Per | 0.605 | **0.725** | −0.003 |
| DahA | 0.164 | 0.199 | 0.507 |
| InP | −0.061 | −0.142 | **0.787** |
| BghiP | −0.029 | −0.085 | **0.744** |
| % of Variance | 46.3 | 16.0 | 10.3 |
| Sources | Coal combustion | Biomass burning | Vehicle exhaust |

Note: r > 0.8 was highlighted in PC1; r > 0.6 was highlighted in PC2 and PC3.

In summary, the sources of PAHs included coal combustion (46.3%), biomass burning (16.0%), and vehicular exhaust (10.3%).

### 3.4.4. Source Identification of n-alkanes by PCA

The source identification of n-alkanes, by diagnostic parameters and $C_{max}$, indicated that biogenic sources contributed the most to n-alkanes. However, this is not enough to infer sources only depending on $C_{max}$ or CPI. For example, hopanes ($C_{31}$) and steranes ($C_{29}$), as tracers of plant wax, have also been found in vehicle exhausts, coal combustion, and ship emissions, vaporized from unburned fossil fuels and lubricant oils. Thus, we conducted PCA analysis for the sources of n-alkanes, and the results are listed in Table 6. Two PCs explained 83.0% of the total variance. PC1 was highly loaded with $C_{26}$–$C_{40}$ n-alkanes, accounting for 59.9% of the total variance. PC2 had strong correlations with medium- and low-molecular-weight n-alkanes ($C_{16}$–$C_{25}$), representing 23.1% of the total variance. According to the previous literature [11], $C_{12}$–$C_{25}$ were derived from fossil fuel burning, including vehicle emissions and coal burning; while $C_{26}$–$C_{36}$ had a biogenic source and were derived from higher plant waxes. Notably, fossil fuel combustion was characterized by the release of $C_{22}$–$C_{25}$ n-alkanes, for example, $C_{21}$ and $C_{25}$ were predominantly emitted from diesel engine exhausts [5].

**Table 6.** Principal component analysis of n-alkanes in $PM_{2.5}$.

| Compounds | PC1 | PC2 |
|:---:|:---:|:---:|
| $C_{15}$ | 0.380 | 0.296 |
| $C_{16}$ | 0.575 | **0.580** |
| $C_{17}$ | 0.658 | **0.578** |
| $C_{18}$ | 0.616 | **0.608** |
| $C_{19}$ | 0.745 | 0.497 |
| $C_{20}$ | 0.730 | **0.588** |
| $C_{21}$ | 0.759 | **0.581** |
| $C_{22}$ | 0.639 | **0.696** |
| $C_{23}$ | 0.598 | **0.689** |
| $C_{24}$ | 0.612 | **0.641** |
| $C_{25}$ | 0.704 | **0.516** |
| $C_{26}$ | **0.801** | 0.333 |
| $C_{27}$ | **0.851** | 0.089 |
| $C_{28}$ | **0.901** | −0.068 |
| $C_{29}$ | 0.454 | −0.315 |
| $C_{30}$ | **0.930** | −0.305 |
| $C_{31}$ | **0.903** | −0.399 |
| $C_{32}$ | **0.897** | −0.426 |
| $C_{33}$ | **0.882** | −0.458 |
| $C_{34}$ | **0.861** | −0.473 |
| $C_{35}$ | **0.871** | −0.470 |

**Table 6.** *Cont.*

| Compounds | PC1 | PC2 |
|---|---|---|
| $C_{36}$ | **0.871** | $-0.462$ |
| $C_{37}$ | **0.872** | $-0.455$ |
| $C_{38}$ | **0.874** | $-0.451$ |
| $C_{39}$ | **0.874** | $-0.439$ |
| $C_{40}$ | **0.884** | $-0.417$ |
| % of Variance | 59.9 | 23.1 |
| Sources | Biogenic source | Vehicle exhaust and coal combustion |

Note: r > 0.8 was highlighted in PC1; r > 0.5 was highlighted in PC2.

Therefore, PC1 was considered to be a contribution of biogenic sources, and PC2 was associated with vehicle exhaust and coal combustion.

### 3.5. Backward Trajectory Analysis for Transport Pattern of Organics

Cluster analysis of backward trajectories was used to determine the transport routes of the air mass. In this study, the 72 h backward trajectories of the air masses arriving at the sampling point at 100 m above ground, at 8:00 local time (LT) during the sampling periods, were analyzed with the hybrid single-particle Lagrangian integrated trajectory (HYSPLIT4.9) model, provided by the NOAA (http://ready.arl.noaa.gov/HYSPLIT.php, accessed on 31 August 2021) global data assimilation system (GDAS).

The results of the cluster analysis are shown in Figure 5. The trajectories can be grouped into four clusters. Among the 75-day sampling period, cluster 1 covers 16 days (severely polluted: 4 days; polluted: 10 days; non-polluted: 2 days); cluster 2 and cluster 3 cover 9 days (severely polluted: 1 day; polluted: 8 days) and 14 days (severely polluted: 1 day; polluted: 13 days), respectively; cluster 4 has 36 days (polluted 17 days and non-polluted 17 days). Overall, cluster 1 (frequency: 21%) originated from Datong (Shanxi Province) and passed through the North China Plain, including some heavily industrialized provinces (Hebei and Shandong), then arrived in Changzhou. Thus, cluster 1 was defined as heavily polluted and a long-range transport air parcel. Cluster 2 (frequency: 13%), derived from air masses originating from Shaanxi Province, passed through Xinyang (Henan Province) and Hefei (Anhui Province), and finally reached the sampling point. Cluster 3 (frequency: 20%) began in Tangshan (Hebei Province), and reached Changzhou after passing across the Bohai Sea and the Yellow Sea. Cluster 4 (frequency: 46%) represented air masses that passed across the East China Sea, and arrived in Changzhou via Shanghai.

For further study, Table 7 listed the concentrations and percentages of PAHs and n-alkanes under four air mass clusters on different polluted days. The concentration of $PM_{2.5}$ was the highest in cluster 2 (126.98 μg/m$^3$), when passing across Henan Province (a heavily industrialized region), while the lowest was in cluster 4 (87.37 μg/m$^3$) from the East China Sea (cluster 4). It can be observed that the concentration of n-alkanes and PAHs did not vary with the same trend of $PM_{2.5}$ among the four clusters. For example, the n-alkane concentration from two sea transports (cluster 3 and cluster 4) was almost equal to that from cluster 2, but the $PM_{2.5}$ concentration was lower than that in cluster 2, due to dilution by clear oceanic air masses. The probable reason for the relatively higher concentration of n-alkanes from the sea was attributed to both the release of n-alkanes from the marine environment and the influence of heterogeneity [54]. Also, some differences were found from the mass distribution of the monomers of n-alkanes and PAHs on different polluted days. In general, the percentages of $C_{26}$–$C_{40}$ (70–90%) were much higher than for $C_{9}$–$C_{25}$ (10–20%). Also, the percentage of $C_{9}$–$C_{25}$ in cluster 3 increased significantly on severely polluted days, suggesting that anthropogenic sources played a larger role. Secondly, six-ring PAHs on severely polluted days in cluster 2 (15.8%) greatly increase compared with other clusters, which implied that vehicle exhaust played a significant role on these days.

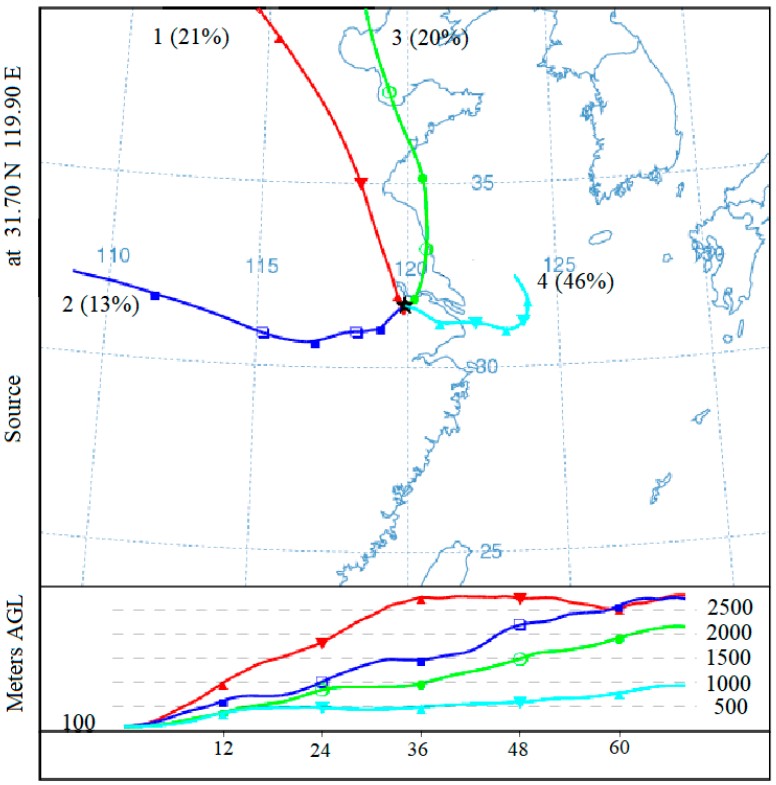

**Figure 5.** Clusters analysis of backward trajectories in 72 h, 100 m during sampling periods at Changzhou.

**Table 7.** Variation in the percentages of n-alkanes and PAHs under four air masses clusters on different polluted days.

| | | | Cluster 1 | Cluster 2 | Cluster 3 | Cluster 4 |
|---|---|---|---|---|---|---|
| | PM$_{2.5}$ (µg/m$^3$) | | 113.63 | 126.98 | 107.69 | 87.37 |
| | $\sum$n-alkanes (ng/m$^3$) | | 316.97 | 264.84 | 208.63 | 237.99 |
| | $\sum$PAHs (ng/m$^3$) | | 5.79 | 3.76 | 4.63 | 3.79 |
| n-alkanes | Severely polluted (%) | C$_9$–C$_{25}$ | 18.61 | 11.64 | 48.17 | 17.39 |
| | | C$_{26}$–C$_{40}$ | 81.39 | 88.36 | 51.83 | 82.61 |
| | Polluted (%) | C$_9$–C$_{25}$ | 28.09 | 12.52 | 12.47 | 19.32 |
| | | C$_{26}$–C$_{40}$ | 71.91 | 87.48 | 87.53 | 80.68 |
| | Non-polluted (%) | C$_9$–C$_{25}$ | 20.74 | - | - | 11.60 |
| | | C$_{26}$–C$_{40}$ | 79.26 | - | - | 88.40 |
| PAHs | Severely polluted (%) | 3-ring | 10.86 | 15.49 | 9.80 | 11.61 |
| | | 4-ring | 54.91 | 30.26 | 55.60 | 62.79 |
| | | 5-ring | 33.99 | 38.41 | 34.54 | 25.22 |
| | | 6-ring | 0.24 | 15.84 | 0.06 | 0.38 |
| | Polluted (%) | 3-ring | 12.08 | 8.86 | 9.01 | 9.02 |
| | | 4-ring | 49.73 | 44.03 | 32.86 | 39.39 |
| | | 5-ring | 34.41 | 43.35 | 55.99 | 50.71 |
| | | 6-ring | 3.78 | 3.75 | 2.15 | 0.88 |
| | Non-polluted (%) | 3-ring | 9.31 | - | - | 11.24 |
| | | 4-ring | 47.67 | - | - | 34.17 |
| | | 5-ring | 42.28 | - | - | 52.39 |
| | | 6-ring | 0.74 | - | - | 2.19 |

## 4. Conclusions

The average concentrations of ∑PAHs and ∑n-alkanes were 4.37 ± 4.95 ng/m$^3$ and 252.37 ± 184.02 ng/m$^3$, respectively, which exhibited higher concentrations on severely polluted days. There was a relative decrease in the percentages of SNA and carbonaceous species, combined with a significant increase in dust in PM$_{2.5}$ on polluted days, implying that the PM$_{2.5}$ concentrations might be somewhat impacted by dust storms.

The relationship between PM$_{2.5}$, ∑n-alkanes, and ∑PAHs was weakly positive. For all the collected PM$_{2.5}$ samples, the average and standard deviation of CPI was 1.35 ± 0.43. A lower CPI value (1.18) under severely polluted days showed more influence to n-alkanes from anthropogenic emissions. The four-to-five-rings PAHs are dominant and occupied over 85% of the total PAHs. The distribution of rings in PAHs also showed that the main sources of PAHs on severely polluted days were coal/biomass combustion, while vehicle emission might have contributed more on non-polluted days. The carcinogenic PAHs species occupied a high proportion (46.3%), implying that more attention should be paid to primary sources, such as vehicle emission and coal combustion. PCA combined with diagnostic ratios revealed that biogenic sources may be relatively more important for PM$_{2.5}$-associated n-alkanes. What is more, the main emission source of PAHs was coal combustion (46.3%), followed by biomass burning (16.0%) and vehicular exhaust (10.3%).

Backward trajectory analysis showed that ∑n-alkanes concentration from two sea transport directions did not decrease with the dilution of clear oceanic air masses, implying that biogenic emission from the sea contributed the most to n-alkanes. Overall, our findings strengthened the fact that regional transport and dust storms in the spring had large impacts on the characterization of PM$_{2.5}$.

**Supplementary Materials:** The following are available online at https://www.mdpi.com/article/10.3390/atmos12091127/s1, Table S1: The explanation of some abbreviations in this study, Table S2: Abbreviations, molecular weight and method detection limits of individual n-alkanes and PAHs in this study, Table S3: Resluts of Mann-Witney statistics between samples in different polluted days.

**Author Contributions:** Methodology, X.L. and H.H.; investigation, Y.J. writing—original draft preparation, N.S.; writing—review and editing, Z.Y. and Z.Z. All authors have read and agreed to the published version of the manuscript.

**Funding:** This work was financially supported by the Natural Science Foundation of Jiangsu Province (BK20181476 and BK20181048), National Natural Science Foundation of China (21976093 and 91544220), Open fund by Jiangsu Key Laboratory of Atmospheric Environment Monitoring and Pollution Control (KHK1904) and the Postgraduate Research & Practice Innovation Program of Jiangsu University of Technolgy (XSJCX20_07).

**Institutional Review Board Statement:** Not applicable.

**Informed Consent Statement:** Not applicable.

**Data Availability Statement:** Not applicable.

**Conflicts of Interest:** The authors declare no conflict of interest.

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
