# Peer review of "Sources of PM2.5-Associated PAHs and n-alkanes in Changzhou China"

_atmosphere, doi:10.3390/atmos12091127_

Round 1

Reviewer 1 Report

This manuscript describes the chemical characterisation of PM2.5 in Changzhou approximately 30 m away from the ground (so we cannot refer to air pollution and effects on human health – The measurements were not performed in breathing zone (2-3m above the ground)). Determined 32 n-alkanes components, 17 kinds of PAHs, OC, EC and some other components in different periods. Also, the possible sources were determined by diagnostic parameters and PCA analysis.

Other minor issues:

You need a table/list of abbreviations or an explanation the first time an abbreviation is used.

Lines 48-52 : In the introduction section, lines 48-52 the authors claimed that n-alkanes have nonreactive characteristics. Where does this refer? N-alkanes are stable compounds, but their reactive.  n-alkanes heterogeneous oxidation lifetime is expected to be on the order of 100 h, when applied to the modelled results (https://doi.org/10.1021/acs.est.5b02115), while on the order of 3-6 days by experimental procedure (https://doi.org/10.3390/atmos8050089).

Line 53 : Mention the authors and not “study [4]”

Line 67: Previous studies [5,13]

Lines 107-108 : change “Tao et al” with “our previous study”.

Lines 123-125 : What method is used to analyse PAHs in PM2.5? The solvent extraction method and manually injected through the GC injector OR using in-injection port thermal desorption gas chromatography/mass spectrometry (TD-GC-MS). Were PAHs measured in the gas or in the particulate phase? A description of the method of PAHs analysis in a few lines is necessary.

Lines 165-166 : The detection limits of the method are calculated statistically through the standards (ISBN-13: 978-0273730422, ISBN-10: 0273730428). Was the MDL set arbitrarily, or is the reference missing? Insert citation.

Line 173 : Add a sentence on how the recovery was calculated

Lines 173-174 : Add a table (either in the main text or as a supplementary material) with MDLs for each PAH and each n-alkane.

Line 177 : It is not a comparison – change the verb (compare) / correct standard deviation

Lines 184-185 : Restatement

Lines 186-189 : Why does the portion of OC and EC in PM2.5 decreased for severely polluted days? Provide an explanation

Lines 189-190 : Perhaps it is good to present the rate for the three pollution stages (fair, polluted and severely polluted).

Lines 196-202 : Show the percentage of n-alkanes and PAHs in PM2.5 for the three pollution stages (fair, polluted and severely polluted). And here it's reduced?

Lines 213-214 : The correlation coefficient is too small to draw any conclusion. Extensive statistical analysis is required. Remove the sentence or rephrase.

Figure 1 : There are days with high PM2.5 concentrations and almost zero PAH concentrations, and the reverse. Why? Explain. Could it be aged pollution? What is the lifetime of PAHs? May also help the percentage of PAHs in PM2.5 per day. Is there a difference when there is a switch from fair to polluted day? / Remove correlation coefficient from the graph.

Paragraph 3.3 : Meteorological conditions can help to determine of source apportionment, and their correlation with n-alkane and PAHs is of no scientific interest. / I think the paragraph should be removed and only the meteorological data should be left, not the correlations.

Line 338 : Insert space between “to” and “1”

Table 3 : Insert the Standard Deviation near the mean value

Table 4 : Rename the title of column 1 from “Scheme 1” to “Cities” or “Sampling Area” / Present the sampling period in the same way in all rows.

Lines 353-354 : Present the equation on a separate line.

Paragraph 3.4.2 : Insert citations where required

Lines 440-441 : Please indicate in detail how many days of each pollution stage each Cluster has.

Reviewer 2 Report

Review

The manuscript by Sun et al. describes the variation in the concentration of n-Alkanes and PAHs concurrently with some other parameters such as OC/EC and some ions.

General comments:

  • The manuscript contains useful data but is focused only on a short time period (spring) and the authors collected 25 samples in March (9 samples), April (8 s) and May (8 s) of 2017. To add value to their work the authors should measure the same parameters during the four seasons (autumn, winter , spring and summer). Twenty five samples only for spring do not represent true time series.
  • The 94% of the citations are citations of Chinese papers. It is certainly author’s choice the papers to cite but then the authors should clarify that the cited papers concern China and no other areas of the world. The later would add value to their discussion, which is not the case in the present manuscript.  

Specific comments:

  1. Title: The title is not appropriate. The authors should avoid the term “different polluted days”. Please find a more appropriate title.

  1. Introduction:

I urge the authors to read and use the following references for:

“Atmospheric particulate … pollution and  adverse effects on human health, visibility and climate”: Seinfeld, J.H., Pandis, S.N., 2016. Atmospheric Chemistry and Physics: From Air Pollution to Climate Change. John Wiley & Sons; Pope, C.A., Dockery, D.W., 2006. Health effects of !ne particulate air pollution: lines that connect. J. Air Waste Manage. Assoc. 56, 709–742; Russell, A.G., Brunekreef, B., 2009. A focus on particulate matter and health. Environmental Science & Technology 43, 4620–4625; Beelen, R., Raaschou-Nielsen, O., Stafoggia, M., Andersen, Z.J., Weinmayr, G., Hoffmann, B., et al., 2014. Effects of long-term exposure to air pollution on natural-cause mortality: an analysis of 22 European cohorts within the multicentre ESCAPE project. Lancet 383, 785–795

“For source reconciliation for n-Alkanes and PAHs”: Gogou et al. “Organic aerosols in Eastern Mediterranean: components source reconciliation by using molecular markers and atmospheric back trajectories”, Organic Geochemistry 25 (1–2), 1996, pp. 79-96; Kavouras et al., “Source Apportionment of Urban Particulate Aliphatic and Polynuclear Aromatic Hydrocarbons (PAHs) Using Multivariate Methods”, Environ. Sci. Technol. 2001, 35, 11, 2288–2294; and others

  1. 46-47: There is no reference that n-alkanes and PAHs are producing SOA. In addition, this sentence contradicts the following in line 52, “n-alkanes provide insight … due to their non-reactive character”.
  2. 51, 56 and below: When you start a sentence you should write n-Alkanes and not n-alkanes. More over when you annotate n-alkanes use the number of carbon atoms as a subscript : e.g. C35
  3. 56-57 and L.62-63: The sentences are not necessary. We know what n-alkanes and PAHs are.

  • Results and discussion:

  1. 196-202 and Table 1: The authors should provide the criterion on which base they classify “severely polluted”, polluted” and “fair” days. They should clearly indicate the basis of which precise parameters (e.g. NOx, PMs, CO, etc.) categorization is done.
  2. 208-244: The discussion is not convincing concerning the differences between samples. The authors are invited to use statistics (e.g. Mann-Witney) to explain the differences between samples for all the parameters they have measured. The term “time series” is exaggerated for only one season and 25 samples.

Section 3.1: In Figure 3 the “different patterns” of n-alkanes distribution are not obvious. The discussion provided between the  L. 258 and L.266 is not convincing. Please consider the references given above.

Section 3.2: The discussion of the results is only descriptive and poorly explained.

Section 3.3: This section doesn’t add a value to the study, especially for PAHs and n-alkanes. The authors did not measure gas phase to deliver a substantial discussion.

Sections 3.4.3&4: Please read your reference 12 and  the paper by Kavouras et al. (“Source Apportionment of Urban Particulate Aliphatic and Polynuclear Aromatic Hydrocarbons (PAHs) Using Multivariate Methods”, Environ. Sci. Technol. 2001, 35, 11, 2288–2294) and rewrite these sections.

Section 3.5: L. 460-467. The conclusions are not well documented and are rather speculative. The authors should indicate on the basis of their results why higher n-alkanes concentrations derive from marine biogenic sources (literature???). There are no monomers of n-alkanes or PAHs. There are homologues of n-alkanes and PAHs members.

Conclusions: L. 481-484. The conclusion is not convincing in relation to the specific section. L. 485-488 Is very speculative.

Round 2

Reviewer 1 Report

Check references (e.g. Do references [11] and [21] refer to what you present in the text?)

Author Response

Response: Thanks for your suggestion. We have changed ref.[11] “Rogge, W.; Hildemann, L.; Mazurek, M.; Cass, G.; Simoneit, B. Sources of fine organic aerosol. 4. Particulate abrasion products from leaf surfaces of urban plants. Environ. Sci. Technol. 1993, 27, 2700-2711”. In addition, we want express “much less attention has been given to compare the characteristics and sources of n-alkanes or PAHs on polluted days with non-polluted days.” The relationship between n-alkanes or PAHs was “or” not “not”, so Ref.21 was appropriate. Moreover, we checked all referenced cited in this paper, and changed ref.41.

Reviewer 2 Report

  • English should be improved. Editing by a specialist is needed.
  • The title is not appropriate. I propose:

“Sources of PM2.5 associated PAHs and n-alkanes in Changzhou China”

  • 44. I disagree with the sentence. PAHs and n-alkanes are not IMPORTANT COMPONENTS OF SOA. The references used do not imply that.
  • I am not in agreement with the use of “different polluted levels” and “fair days”. The authors should use more appropriate terminology.
  • Table 1: They should write “constituents” and not “constitutes”
  • The authors use past tense in a wrong way: e.g., “Time series were showed” instead of “Time series are shown”.
  • 1: I am not in favor to use the term “time series” for a period of 3 months and 24 samples.
  • I still have the same concerns for the conclusions as expressed in my previous review.

Author Response

1.English should be improved. Editing by a specialist is needed. The authors use past tense in a wrong way: e.g., “Time series were showed” instead of “Time series are shown”.

Response: Thanks. We have paid more attention to the correct use of tense for the whole paper. English has been improved.

  1. The title is not appropriate. I propose:“Sources of PM2.5 associated PAHs and n-alkanes in Changzhou China”

Response: Thanks. The title has been changed to “Sources of PM2.5 associated PAHs and n-alkanes in Changzhou China”.

  1. I disagree with the sentence. PAHs and n-alkanes are not IMPORTANT COMPONENTS OF SOA. The references used do not imply that.

Response: Thanks for your suggestion. Since we found that a series of paper focused on aqueous-phase reaction and gas phase photochemical oxidation for our target precursor, we mistook their meaning. Actually, they wanted to express that IVOC were a class of important SOA precursors. Anyway, we also agreed with you that PAHs and n-alkanes are not important components of SOA, so we deleted this sentence.

  1. I am not in agreement with the use of “different polluted levels” and “fair days”. The authors should use more appropriate terminology.

Response: Thanks. We have changed “fair days” to “non-polluted days”

  1. Table 1: They should write “constituents” and not “constitutes”

Response: Thanks. Done.

  1. I am not in favor to use the term “time series” for a period of 3 months and 24 samples.

Response: Sorry for that. But I have to again give an explanation that we analyzed 75 samples not 24 samples. So, we think the term “time series” was appropriate.

  1. I still have the same concerns for the conclusions as expressed in my previous review.

Response: Thanks. We have rewritten Section 4 and think that conclusion is now convincing. For example, the first paragraph has been changed to “The average concentration of ∑PAHs and ∑n-alkanes were 4.37±4.95 ng/m3 and 252.37±184.02 ng/m3, which exhibited higher concentration on severely polluted days. Relatively decrease in percentages of SNA and carbonaceous species, combined with significant increase of dust in PM2.5 in polluted days, implying that the PM2.5 concentrations might be somewhat impacted by dust storms.”.
